# Assessment of Nasal Structure Using CT Imaging of Brachycephalic Dog Breeds

**DOI:** 10.3390/ani12131636

**Published:** 2022-06-25

**Authors:** Ryo Oshita, Sakie Katayose, Eiichi Kanai, Satoshi Takagi

**Affiliations:** Laboratory of Small Animal Surgery, School of Veterinary Medicine, Azabu University, Sagamihara 252-5201, Japan; dv2105@azabu-u.ac.jp (R.O.); katayosesakie@gmail.com (S.K.); kanai@azabu-u.ac.jp (E.K.)

**Keywords:** dog, respiratory, brachycephalic airway syndrome, computed tomography, nasal cavity

## Abstract

**Simple Summary:**

French bulldogs and pugs are classified as brachycephalic dogs and are characterized by short, stenotic noses. The outward characteristics of this breed can cause dyspnea, which may be surgically improved. However, brachycephalic dog breeds have different rates of disease progression and response to treatment. We hypothesized that this difference was due to the difference in the structure of the nasal cavity. To verify this, the ratio of the airway cross-sectional area to the total nasal cavity area was compared between the three brachycephalic dog breeds using head computed tomography (CT) images. The results revealed that the distribution of narrow and wide airway locations in the nasal cavity differed among the breeds. The method used in this study was suitable for observing and evaluating canine nasal structures. These results provide a new approach for the diagnosis of canine respiratory diseases and improve our understanding of the pathogenesis of brachycephalic dogs.

**Abstract:**

The response to treatment of brachycephalic airway syndrome (BAS) varies among brachycephalic dog breeds. We hypothesized that variations in nasal structure are one of the factors responsible for this difference. To confirm this variation, we measured the ratio of the airway cross-sectional area to the total nasal cavity area (AA/NC) in three brachycephalic dog breeds. Head CT images of French bulldogs, shih tzus, and pugs were retrospectively collected. Four specific transverse planes were used to calculate AA/NC ratios. Fifty brachycephalic dogs were included in the study: French bulldogs (*n* = 20), shih tzus (*n* = 20), and pugs (*n* = 10). The AA/NC ratio of Shih Tzus was larger in the rostral nasal cavity and smaller toward the caudal area, whereas the other two breeds showed an inverse tendency. The results obtained from the current research indicate that the AA/NC ratio can be used to evaluate the structure of the nasal cavity. Moreover, analyzing the point with the smallest AA/NC ratio can be useful in quantifying nasal airway obstruction and the severity of BAS. These results will be useful in understanding the complexity of BAS pathophysiology and in the implementation of treatment.

## 1. Introduction

Brachycephalic dogs have a short nose and wide skull, which, along with related deformities, and are known to cause difficulty breathing [1,2]. This breed-specific airway obstruction disease is recognized as brachycephalic airway syndrome (BAS) and is characterized by progressive disease [3,4,5,6]. Breeding selection for extreme brachycephalia has resulted in deformation of the upper airway tract, leading to obstruction and formation of negative pressure, as the soft tissues have not reduced proportionately with the length of the skull [7]. Symptoms of this deformation include combinations of snoring, coughing, abnormal respiratory sounds, dyspnea, regurgitation, vomiting, exercise intolerance, cyanosis, and syncope [3,8,9,10,11]. Negative pressure caused by narrowing of the oronasal airway may induce collapse of airway structures, resulting in a variety of secondary problems other than upper airway obstruction, such as gastrointestinal disorders and pulmonary edema [3,5,12,13]. BAS should therefore be treated as early in life as possible [14,15].

Typical anatomical abnormalities that cause this syndrome are a hyperplastic soft palate, stenotic nares, and eversion of the laryngeal saccules. According to previous reports, the incidence rate of each abnormality in brachycephalic breeds ranges from 85.6–94% for hyperplastic soft palate, 42.5–77% for stenotic nares, and 58.9–66% for eversion of the laryngeal saccules [16,17,18,19]. In many brachycephalic dogs, a combination of these abnormalities often occurs [17,18,19]. Surgery is the standard treatment for these conditions. Routine surgical treatment includes correction of stenotic nares, reduction of the soft palate, and removal of the laryngeal saccules by removing residual tissues from each part [18]. Many brachycephalic dogs show improved clinical signs after surgery; however, 50% of dogs remain clinically ill despite amelioration of their respiratory problems [5,16,18,19]. Hence, clarification of the exact pathophysiology of BAS is necessary.

A number of studies based on computed tomography (CT) images of dogs with BAS have reported the phenomenon of ectopic tissue protruding into the nasal cavity because it cannot be accommodated by the internal structures owing to shortening of the skull [20,21,22,23]. A recent prospective clinical study reported that 66.7% of brachycephalic dogs that underwent head CT scans had aberrant turbinates, and 91.7% of the dogs were found to have interconchal and intraconchal mucosal contacts. That report discussed the presence of abnormal structures in the nasal cavity as a possible cause of inadequate results with conventional surgery [23].

There are more than 10 breeds that conform to the definition of brachycephalic dog. In studies on BAS from 2005 to 2010, English bulldogs and pugs accounted for approximately 25%, and French bulldogs accounted for approximately 20%; in contrast, shih tzus accounted for only 3% of the total [5,6,20,22,24,25]. Surgical treatment of shih tzus with stenotic nares was reported to be excellent, and the clinical signs in all dogs improved [26]. Thus, it is suggested that even brachycephalic dogs with similar muzzle lengths show different responses to treatment.

Considering this background, we hypothesized that the intranasal structure may differ even among brachycephalic dog breeds. Therefore, the purpose of this study was to investigate the differences in nasal structure among brachycephalic dog breeds by measuring the ratio of the airway cross-sectional area to the total nasal cavity area (AA/NC).

## 2. Materials and Methods

### 2.1. Animals

Medical records of dogs that underwent abdominal contrast-enhanced head CT imaging under general anesthesia at Hokkaido University Veterinary Teaching Hospital between October 2014 and January 2018 and Azabu University Veterinary Teaching Hospital between November 2017 and April 2021 were examined; French bulldogs, shih tzus, and pugs were included in the study.

CT images were obtained using an 80-row multi-slice CT scanner (Aquilion PRIME; Toshiba Medical System, Tochigi, Japan). The studies were performed in the transverse plane using a bone window. Slice thickness varied between 0.5 mm and 3.0 mm.

### 2.2. Exclusion Criteria

Cases with intranasal space-occupying lesions or lesions infiltrating the nasal cavity were excluded. Cases with perinasal bone defects due to diseases other than neoplasms were also excluded. Likewise, cases without the cross-sectional CT images necessary for analysis or with poorly positioned images were excluded.

### 2.3. Measurement Method

#### 2.3.1. 2D Region Growing

The CT images were displayed and analyzed using OsiriX DICOM viewer (Pixmeo SARL, Geneva, Switzerland). In this study, we used a 2D region-growing method that automatically extracts all pixels on the screen and sets them as the region of interest (ROI) by establishing a pixel threshold for a specific region of the CT image [27].

#### 2.3.2. Measurement of the AA/NC Ratio

Four planes were selected for the analyses: first incisor teeth level (middle section showing maximum alveolar bone area of the first incisor tooth), canine teeth level (middle section showing maximum alveolar bone area of the canine tooth), palatal horizontal plate level (the most rostral cross section showing the palatine horizontal plate), and suborbital foramen level (the most caudal cross section showing both suborbital foramina). The cross-sectional areas of the airway and nasal periphery in these four selected planes were measured using OsiriX DICOM viewer, and the AA/NC ratio of the areas was compared between breeds (Figure 1).

The cross-sectional area of the airway was measured using 2D region growing with a threshold set at a CT value of −1024 to −450 HU [28]. The cross-sectional area of the perinasal cavity was measured using 2D region growing with a threshold set at a CT value of −1024 to −150 HU [29]. If the ROI of the nasal cavities was not surrounded by bone, CT values could not be applied; thus, the area was measured by manually surrounding the nasal cavity using a freehand ROI tool. If the cross-sectional images were not symmetrical, 3D multiplanar reconstruction was used to adjust the angle of the slice for measurement, and the ROI was set with reference to the shape of the bone in a nearby cross-section of the section to be measured. The AA/NC ratio was calculated by dividing the cross-sectional area of the airway by the total area of the nasal cavity.

### 2.4. Statistical Analyses

All statistical analyses were performed using JMP 14 SW (JMP Japan, Tokyo, Japan). The significance level was set at 0.05 for all tests, unless otherwise indicated.

The normality of the distribution of the AA/NC ratio measured in the three breeds was tested using the Shapiro–Wilk test. The test for multiple comparisons was modified based on whether measurements in the three breeds were normally distributed. If the measurements in the three breeds were normally distributed, one-way analysis of variance (ANOVA) was performed. Multiple comparisons were performed using Tukey–Kramer’s HSD test only if there was a significant difference between the three groups. If the measurements in the three breeds were not normally distributed, the Kruskal–Wallis test was performed. Multiple comparisons were performed using Steel–Dwass’s test only if there was a significant difference between the three groups.

## 3. Results

### 3.1. Study Population

Sixty-two dogs met the inclusion criteria for this study: French bulldogs (*n* = 23), shih tzus (*n* = 24), and pugs (*n* = 15). Of these, 12 animals that did not meet the criteria were excluded. The reasons for exclusion were lack of measurement cross-section (*n* = 6), osteolysis due to tumor invasion (*n* = 1), tumor existence or invasion into the nasal cavity (*n* = 4), and loss of palatine bone due to cleft palate (*n* = 1).

Therefore, 50 dogs were included in this study. The dog breed distributions were as follows: French bulldogs (*n* = 20), shih tzus (*n* = 20), and pugs (*n* = 10). The median age of each dog breed was 8 (range, 1–13) years, 10.5 (6–13) years, and 8 (2–13) years, respectively. The gender distribution of each dog breed was 10 males (9 castrated) and 10 spayed females, 14 males (7 castrated, 4 unknown) and 6 females (4 spayed, 1 unknown), and 8 males (6 castrated, 1 unknown) and 2 spayed females, respectively.

### 3.2. The AA/NC Ratio

Figure 2 illustrates the results of the comparison between the AA/NC ratios of the three breeds for the four planes. At the first incisor and canine tooth levels, the area ratio for shih tzus was significantly greater than that for the other two breeds (*p* < 0.05) (Figure 3). On the other hand, at the palatal horizontal plate level, the area ratio for shih tzus was significantly smaller (*p* < 0.05) than that for the other breeds. At the suborbital foramen level, the area ratio for the French bulldogs was significantly greater than that for the other two breeds (*p* < 0.05) (Figure 4).

Moreover, French bulldogs and pugs showed a tendency for the AA/NC ratio to increase from the rostral to the caudal direction. In contrast, the area ratio for the shih tzus was highest at the canine teeth level and lowest at the palatal horizontal plate level (Table 1).

French bulldogs and pugs showed a tendency for the AA/NC ratio to increase from the rostral to the caudal direction. In contrast, the area ratio for the shih tzus decreased from the rostral to the caudal side.

## 4. Discussion

The purpose of this study was to investigate the differences in nasal structure among brachycephalic dog breeds by measuring the AA/NC ratio. Interestingly, breeds with severe respiratory symptoms caused by BAS showed a tendency for the AA/NC ratio to increase from the rostral to the caudal direction. On the other hand, breeds that were deemed not susceptible to BAS in the researchers’ experience showed a tendency for the ratio to be smaller on the caudal side. In particular, shih tzus were infrequently included in BAS studies and data were insufficient. One of the novelties of this study is the detailed observation of the nasal structure of the shih tzu, which empirically has milder respiratory symptoms from BAS.

At the first incisor and canine tooth levels, the area ratio was significantly smaller for French bulldogs and pugs than for shih tzus. Auger et al. (2016) conducted a study that compared the percentage of nasal mucosal contact using CT images in brachycephalic and normocephalic dogs. In that report, brachycephalic dog breeds had more mucosal contact between the concha nasalis media and concha nasalis ventralis than normocephalic dog breeds [22]. However, they did not mention differences in intranasal structure among brachycephalic dog breeds. Although mucocutaneous contact was not measured in this study, it is possible that breeds with smaller AA/NC ratios have relatively larger mucocutaneous contact areas. In a study using CT images and computational fluid dynamics of the English bulldog, Hostnik et al. (2017) showed that the rostral one-third of the nasal passage exhibited a larger airway resistance than the caudal and middle regions of the nasal passage [28]. Those results and the results of our study support the hypothesis that stenosis in the rostral nasal cavity is a significant risk factor for BAS severity. However, Hostnik et al. (2017) noted that the measurement variability of the quantified resistance was large among their 21 dogs [28]. The AA/NC ratio used in our study could not be used to directly evaluate the location of airflow resistance. Therefore, the correlation between the AA/NC ratio and the results of hydrodynamic tests should be examined.

Measurement of the AA/NC ratio using 2D region growing is a useful method for observing and evaluating the structure of the dog’s nasal cavity. Preliminary manual measurements were initially performed and compared with automatically obtained data using the 2D growing method, with a highly reproducible outcome (data not shown). Therefore, in this study, all the regions of interest were set up for 2D growth. This method can facilitate analyses using objective, simple, and quick measurements.

This study has several limitations. First, data were retrospectively collected from multiple centers; thus, the presence of clinical signs, clinical grade of the BAS, and scanning settings, such as slice thickness, were not available in some cases. Therefore, the relationship between the AA/NC ratio and the severity of clinical signs could not be determined. In the BAS grading report, grading can be categorized according to the frequency of clinical signs [12] and the degree of intolerance after exercise testing [30]. A prospective blinded study is needed to evaluate the significance of the correlation between the AA/NC ratio and the BAS grading system.

In addition, the retrospective study design affected the size of the sample. The study included 50 cases that fit the established inclusion criteria. We believe this sample size was sufficient to test the hypotheses established in this study. However, the nasal structure of brachycephalic dog breeds is very varied, and the possibility that the relatively small number of samples influenced the statistical results cannot be ruled out.

In this study, the specific intracranial structures were used to set up the cross-section to be measured. However, the length of the nasal cavity varies slightly between samples; thus, the designated cross-sections do not necessarily show the same coordinates in the nasal cavity. Further sample accumulation is needed to accurately assess the increase or decrease in nasal area.

Opportunities for CT imaging of the head are limited, and CT studies are rarely used in routine practice for the diagnosis of BAS. In this study, we were able to collect many CT head images of brachycephalic dog breeds, but many cases were excluded from the sample according to the set criteria because they were examined for a different reason than for BAS (for example, nasal or oral tumors, dental disease). In order to identify more detailed trends in nasal structures in brachycephalic dog breeds, we believe that head CT images should be actively collected for the purpose of diagnosing BAS.

Another limitation was the lack of skull index data. In a study using head CT images of dogs, normalization of measurements between breeds is commonly performed using this index, which is defined as (skull width)/(skull length) × 100 [20,22,31]. However, the skull index was not available in some cases because of an insufficient scan range. It is suggested that area–ratio comparisons can provide information for assessing the nasal structure of brachycephalic dogs without using normalization. The skull index is suitable for external characterization of cranial features. However, it does not provide an accurate picture of the structures within the nasal cavity. In the future, additional research is required to investigate whether there is a difference in accuracy between the evaluation of the area ratios and the measurement.

Age was not included as an exclusion criterion. To the best of our knowledge, there are no reports on the effects of aging on the nasal structures of dogs. However, in humans, it has been suggested that the nasal airways are wider in seniors than in youths. The main reasons for the structural differences with age are the decrease in the nasal mucosa and collapse of the intranasal structure due to aging [32]. Therefore, it may be necessary to consider possible age-related changes when evaluating a dog’s nasal cavity.

## 5. Conclusions

It is possible to evaluate the nasal cavity structure in dogs using the AA/NC ratio. Moreover, analyzing the point with the smallest AA/NC ratio may be useful in quantifying nasal airway obstruction and BAS severity.

In the future, research should focus on the relationship between BAS severity, degree of improvement due to surgery, and the AA/NC ratio.

## Figures and Tables

**Figure 1 animals-12-01636-f001:**
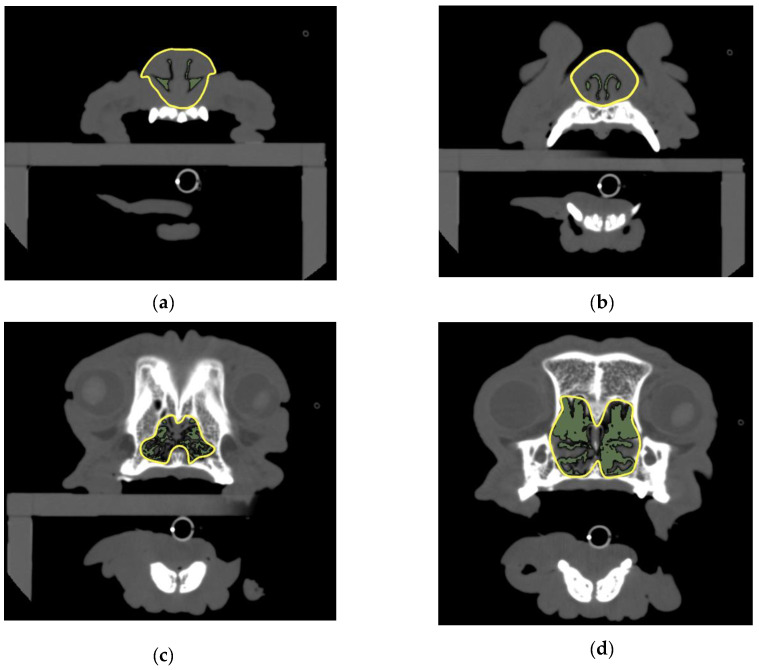
Measurements of the AA/NC ratio.(**a**) Cross section showing alveolar bone of first incisor tooth; (**b**) canine tooth level: cross section showing alveolar bone of canine tooth; (**c**) palatal horizontal plate level: the most rostral cross section showing the palatine horizontal plate; (**d**) suborbital foramen level: the most caudal cross section showing both suborbital foramina. AA/NC, airway cross-sectional area/total nasal cavity area.

**Figure 2 animals-12-01636-f002:**
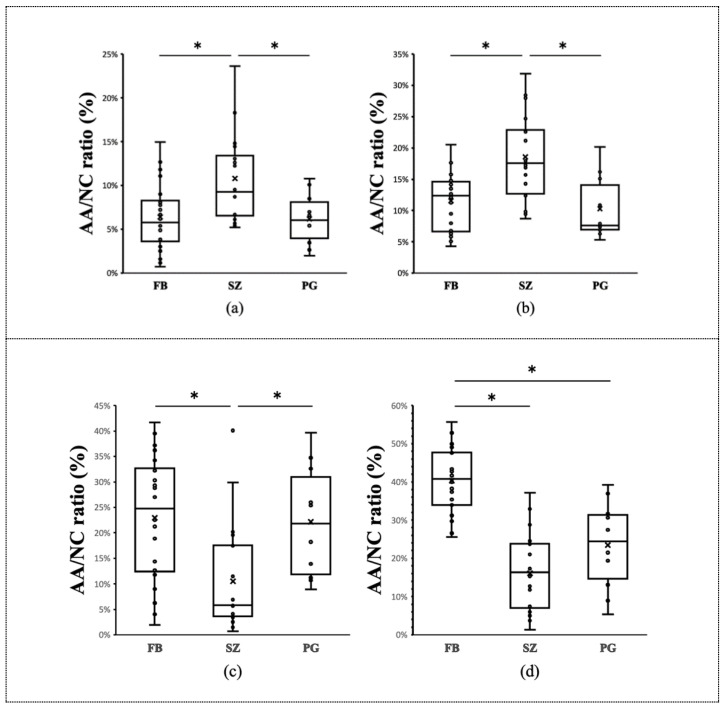
Comparison of the AA/NC ratio among three brachycephalic dog breeds. Comparison of the AA/NC ratio among three dog breeds (French bulldogs, FB; shih tzus, SZ; pugs, PG) using four cross sections. (**a**) First incisor tooth level; (**b**) canine tooth level; (**c**) palatal horizontal plate level; (**d**) suborbital foramen level. At the first incisor and canine tooth levels, the AA/NC ratio for the shih tzus was significantly greater than that for the other two breeds (*p* < 0.05). At the palatal horizontal plate level, the AA/NC ratio for shih tzus was significantly smaller (*p* < 0.05) than those for the other breeds. AA/NC, airway cross-sectional area/total nasal cavity area (*: *p* < 0.05).

**Figure 3 animals-12-01636-f003:**
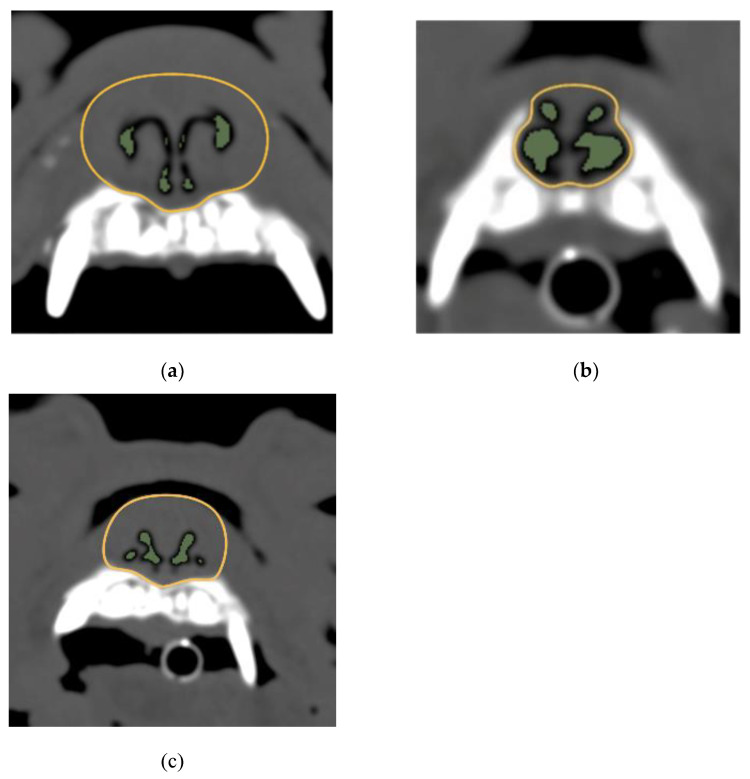
Images of canine teeth levels and the AA/NC ratios for three dog breeds. (**a**) French bulldog; (**b**) shih tzu; (**c**) pug. The following are typical images and measurements. The AA/NC ratio was the highest in Shih Tzus (27.98%). There were no clear differences between the French bulldogs (5.07%) and pugs (7.05%). AA/NC, airway cross-sectional area/total nasal cavity area.

**Figure 4 animals-12-01636-f004:**
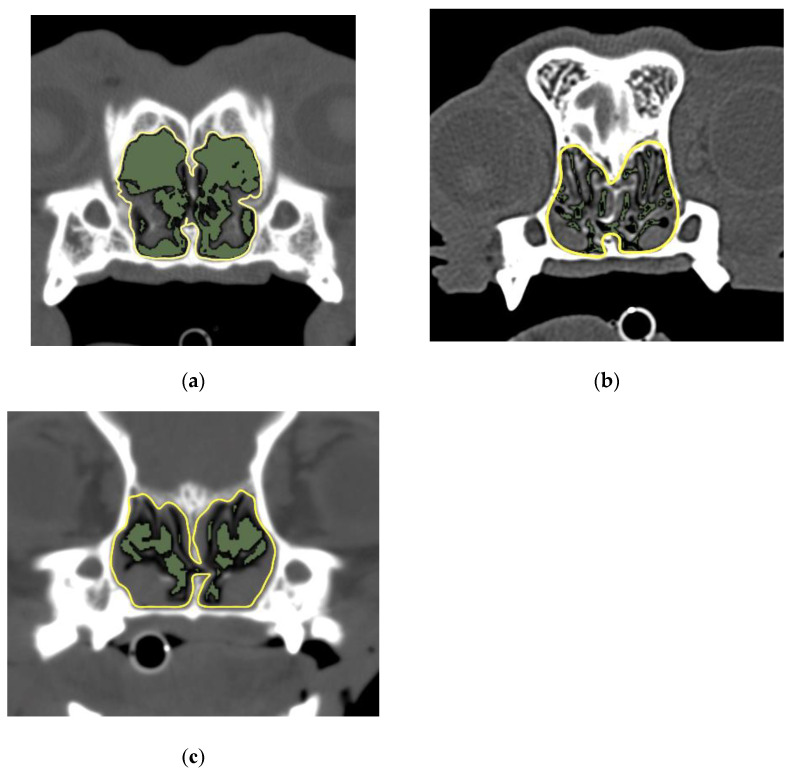
Image of suborbital foramen level and the AA/NC ratios for three dog breeds. (**a**) French bulldog; (**b**) shih tzu; (**c**) pug. The AA/NC ratio was the highest in French bulldogs (55.54%). The following are the typical images and measurements. There were no clear differences between shih tzus (21.03%) and pugs (21.47%). AA/NC, airway cross-sectional area/total nasal cavity area.

**Table 1 animals-12-01636-t001:** Median AA/NC ratios in four planes (%).

	FB	SZ	PG
**First incisor teeth level**	5.79	9.26	6.06
**Canine teeth level**	12.38	17.58	7.63
**Palatal horizontal plate level**	24.77	5.84	21.80
**Suborbital foramen level**	40.82	16.30	24.44

FB, French bulldogs; SZ, shih tzus; PG, pugs; AA/NC, airway cross-sectional area/total nasal cavity area.

## Data Availability

Not applicable.

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
