# Peer review of "Assessment of Nasal Structure Using CT Imaging of Brachycephalic Dog Breeds"

_animals, 2022, doi:10.3390/ani12131636_

Round 1
Reviewer 1 Report
The purpose of this study was to investigate the differences in nasal structure among brachycephalic dog breeds by measuring the ratio of the cross-sectional area airways in relation to the total nasal cavity area (AA/NC). In my opinion the low number of animals (small sample size) employed for every breed may not be statistically relevant for the average of those breeds.
The study is original in nature and well-designed, but maybe not overly relevant in that it really doesn´t reveal important new information on the morphology of these kinds of breeds.
I have a couple of reservations: one, the sample size of only 10 Pugs seems to be overly small to be highly scientifically relevant for this breed; and second, while the 4 images are well done, they do not seem to reveal much of anything new or unexpected which would contribute much to our understanding of the nature of the problems associated with brachycephalic breeds. Such limitations are detailed in lines 235-257 by the authors and I agree with.
They need to standardize their methodology concerning how they take measurements of the area, and they should SIGNIFICANTLY INCREASE the number of examined animals in each breed.
In addition, in the References section I have found some typographical errors which need to be corrected.
Ref 21 Vilaplana Grosso F, instead of Vilaplana G.; Federico.
JAAHA, probably J Am Anim Hosp Assoc according to other journals
Referecence 9. J Small Anim Pract. 1993.34, 515-519
Author Response
Please see the attachment.
Thank you very much for your peer review.

Reviewer 2 Report
Interesting study and well written. My main concern is methodology of using teeth to mark areas in the nasal cavity and whether this means that you are not measuring the same areas of the nasal cavity across the breeds depending on how the nasal cavity is positioned. Please see file.

Author Response

(The authors gave the same response as above.)

Reviewer 3 Report
It is a well-written manuscript, with an innovative approach to the pathogenesis of the brachycephalic airway syndrome (BAS) and helps to explain the differences in symptomatology and treatment outcome between the brachycephalic breeds studied. These findings identify the narrowest area of the nasal cavity, which are the critical points, where action must be taken to alleviate the BAS. However, it would have been a more comprehensive study if it would have been complemented with other techniques, such as the computational fluid dynamics.
The main advantage of this research is its simplicity. Having the nasal cavity of brachiocephalic breeds characterized, would help to decide the most appropriate treatment of BAS according to the dog breed, depending on whether the narrowing of the nasal cavity is anterior or posterior.
Trying to characterize the syndrome, other research articles (listed the references as):
- [22] used CT in brachyocephalic and normocephalic dogs to measure the angle of septal deviation, to confirm the presence of caudal aberrant nasal conchae, in the nasopharyngeal meatus and in choanae; and quantification of nasal mucosal contacts at different anatomical locations. These authors did not allude the differences in the nasal structure between the seven different brachyocephalic breeds in their study (n=18, although not including the Pugs), issue that has been properly addressed in this manuscript with a comparative approach between the three breeds studied and with a larger number of animals.
- [23] used a bigger sample (n=132), in contrast to the 50 dogs that met the inclusion criteria in this manuscript (n=20 to each French bulldog and Shih Tzu groups, and n=10 to Pug group). In this article all dogs had abnormal conchal growth that obstructed the intranasal airways: rostral aberrant turbinates (RAT) and caudal aberrant turbinates (CAT) obstructing the nasopharyngeal meatus. They also studied the deviation of the nasal septum. In this study, two brachyocephalic dog breeds studied were the same as those that appear in the present manuscript: French bulldog and common Pugs, and one different: English bulldog, instead of Shih Tzu of this manuscript. However, in this manuscript, cases with intranasal space-occupying lesions or lesions infiltrating the nasal cavity were excluded. Hence, the samples (regardless of the number) are different.
- [28] used CT and Computational fluid dynamics to quantify and characterize airway resistance through the nasal cavity in English bulldogs, but not in any of the three breeds studied in the present manuscript. They got a 3D model of nasal passage, front sinus and nasopharyngeal passage. Computational fluid dynamics models derived from nasal multidetector computed tomography can quantify airway resistance in brachycephalic dogs, however the process to generate tetrahedral mesh elements used in the computational fluid dynamics analysis in a three-dimensional model from CT images of the nasal passages was not simple and complex software is required. In the present manuscript, the frontal sinus was not the object of study, and only the nasal passages and nasopharynx were considered. In addition, CT scanning at four levels in the nasal cavity is enough to establish a model of the nasal cavity, locating its widest part and where it narrows the most in the three different breeds.
-In [28], most of the English bulldogs studied exhibited a larger airflow resistance in the rostral third of the nasal passage than in the caudal and middle regions of the nasal cavity, results that could be correlated to those obtained in this manuscript on French bulldogs and Pugs, as they showed a tendency to increase the airway area/total nasal cavity area (AA/NC) from rostral to caudal.
Initially, the use of these complex methods is recommended to establish correlations with other simpler techniques, so that, once the correlations are stablished, the procedure to obtain the required information can be simplified.
I think that the methodology is appropriate to the study and to reach a general approach of the morphology of the nasal cavity in these three dog breeds studied, in order to predict in each breed the representative location of the intranasal narrowings.
The conclusions are consistent with the results obtained with the methodology used (CT of the nasal cavity at four levels), obtaining a general pattern of the nasal cavity morphology in each breed dog studied, useful to get a deeper insight into the brachyocephalic airway syndrome (BAS).
All the listed references are appropriate to the topic discussed. Although there are two minor errors:
-[21] This article (DOI: 10.1111/vru.12249) in the text has these authors: Vilaplana, G.; Federico, G.T.H; Susanne, A.E.B. In the two last authors, their surname has been
Two minor comments:
-In the Results chapter, the paragraph L168-171 is repeated in L201-203
-L221 and 225: the publication year is missing after the author’s name: Hostnik (2017)
Author Response
Thank you very much for your peer review.
Please see the attachment.

Reviewer 4 Report
Interesting publication, the idea of using CT scans to develop an AA / NC ratio index worth showing to a wider group of veterinarians.
The most valuable part of this publication is the clear definition of the limitations associated with the radiological diagnosis of brachycephalic dogs.
For my part, I would like to add that CT examination is not yet widely available and relatively expensive, for example compared to traditional X-ray.
Another obstacle in the widespread use of CT in the veterinary medicine of small animals are the contraindications to anesthesia - especially in the course of severe infections of the upper and lower respiratory tract.
Therefore, the development of the AA / NC ratio index is worth presenting to a wider group of recipients as an additional diagnostic tool and possibly the profiling of prophylactic treatment for short-haired dogs.
Additional opinion:
1. The research hypothesis put forward by the authors is based on the assumption that the different structure of the nasal cavity in brachycephalic times is the reason for the differentiation of clinical symptoms and treatment effects compared to mesocephalic dog breeds.
The assumption is formulated in an understandable and logical way.
2. The topic may not be very original, but in my opinion, it requires additional data, information that can be read from radiographs or computed tomography scans, fit into the area of the so-called basic research and should be performed whenever possible. Therefore, the subject of creating a new craniometric index based on CT scans is important and useful in diagnostic practice, especially in dog breeds predisposed to a specific group of diseases (in this case, the upper and lower respiratory tract).
3. In my opinion, a clear sentence should be added that CT examinations of the nasal cavity were performed on the occasion of other clinical symptoms that indicated an indication for CT examination.
It is not common practice in small animal veterinary medicine to refer a patient for a CT scan for respiratory symptoms.
4. The conclusions are very general, but on the basis of the described methodology and the results obtained - it would be excessive to draw too bold conclusions.
Round 2
Reviewer 1 Report
Accepted the changes and comments of the authors.